# The Diagnostic Challenge of a False-Positive Cryptococcal Antigen in Chronic Meningitis with Suspected Indolent CNS B-Cell Lymphoproliferative Neoplasm

**DOI:** 10.3390/jof11100697

**Published:** 2025-09-25

**Authors:** MohammadReza Rahimi Shahmirzadi, Melissa Fowler, Lise Bondy, Seth Climans, Jonathan Lau, Eric To, Yiannis Iordanous, Marilyn Phung, Fatimah AlMutawa, Jeff Fuller, Michael Silverman

**Affiliations:** 1Division of Infectious Diseases, Western University, London, ON N6A 4V2, Canada; melissa.fowler@vch.ca (M.F.); michael.silverman@sjhc.london.on.ca (M.S.); 2Department of Clinical Neurological Sciences, Western University, London, ON N6A 5W9, Canadajonathan.lau@lhsc.on.ca (J.L.); 3Division of Hematology and Oncology, Western University, London, ON N6A 5W9, Canada; eric.to@lhsc.on.ca; 4Department of Ophthalmology, Western University, London, ON N6A 4V2, Canada; yiannis.iordanous@sjhc.london.on.ca; 5Department of Internal Medicine, Western University, London, ON N6A 5W9, Canada; marilyn.phung@lhsc.on.ca; 6Department of Pathology and Laboratory Medicine, Western University, London, ON N6A 5W9, Canada; fatimah.almutawa@lhsc.on.ca (F.A.); jeffrey.fuller@lhsc.on.ca (J.F.); 7Department of Microbiology and Immunology, Western University, London, ON N6A 3K7, Canada

**Keywords:** meningitis, *Cryptococcus*, B-cell lymphoproliferative neoplasm, case report, false-positive

## Abstract

A 47-year-old woman presented with a two-year history of progressive visual symptoms and headaches. Lumbar puncture revealed lymphocytic pleocytosis, elevated protein, low glucose, and a CSF CrAg titer of 1:256. She was treated empirically for cryptococcal meningitis with amphotericin B, flucytosine, and fluconazole for 15 months. Her symptoms persisted, and repeated CSF and serum CrAg, fungal cultures, and an extensive infectious workup were negative. CSF flow cytometry eventually demonstrated a monoclonal B-cell population suggestive of a lymphoproliferative process. Imaging, including MRI and PET scans, did not reveal systemic disease. A ventriculoperitoneal (VP) shunt was placed for symptom management. This case emphasizes the limitations of CrAg testing and the potential for false positives. It underscores the need for integrating clinical, laboratory, and imaging data when evaluating chronic meningitis.

## 1. Introduction

Meningitis is a potentially life-threatening syndrome characterized by inflammation of the meninges, which may present acutely or chronically depending on the underlying etiology. Acute meningitis is most often viral or bacterial, while chronic meningitis defined as clinical meningitis with cerebrospinal fluid (CSF) pleocytosis lasting more than four weeks has a broad differential diagnosis [1,2]. Causes include infections (fungal, mycobacterial, parasitic, atypical bacterial), inflammatory and autoimmune disorders (neurosarcoidosis, CNS vasculitis, IgG4-related disease, systemic lupus erythematosus), and malignancy [1,2]. Among infectious causes, *Cryptococcus* species are the most common fungal pathogens leading to chronic meningitis. *Cryptococcus neoformans* primarily affects immunocompromised hosts, particularly those with advanced HIV infection, while *Cryptococcus gattii* can also cause disease in immunocompetent individuals [3,4,5]. Diagnosis relies on clinical suspicion, CSF findings, culture, and cryptococcal antigen (CrAg) testing. CrAg assays including latex agglutination, enzyme immunoassay, and lateral flow assays are highly sensitive and specific, and their rapid turnaround has made them indispensable for early recognition of cryptococcosis. In HIV-infected populations, CrAg testing has shown excellent diagnostic performance, with sensitivity and specificity exceeding 98% in both serum and CSF [6]. In this setting, a positive CrAg is highly predictive of true infection and typically warrants immediate antifungal therapy. By contrast, in immunocompetent or non-HIV patients, the interpretation of CrAg positivity is more challenging because the pretest probability of cryptococcosis is lower and the differential diagnosis broader [7,8]. False-positive CrAg results, although uncommon, have been described in multiple contexts. Cross-reactivity with other fungi, such as *Trichosporon beigelii*, *Schizophyllum commune*, *Alternaria alternata*, and *Mucor circinelloides*, has been reported [9].

Interference from immunoglobulins, particularly macroglobulins and rheumatoid factor, can produce nonspecific agglutination, with cases described in systemic lupus erythematosus, rheumatoid arthritis, and hematologic malignancies [10,11]. High titers have occasionally been documented in the absence of cryptococcal infection, particularly in patients with hematologic disease [10]. Technical issues, including contamination and cross-reactivity with specimen transport media, have also been implicated [12]. In addition, the prozone effect where excessive antigen concentrations inhibit visible agglutination can result in paradoxical false negatives unless serial dilutions are performed [8]. Reported frequencies of false positivity are generally low in HIV-infected patients but appear higher in immunocompetent individuals or those with alternative pathology. This variability underscores the importance of interpreting CrAg results in a clinical context. In the case we present, an immunocompetent woman with chronic meningitis was initially diagnosed with cryptococcal meningitis on the basis of a high CSF CrAg titer but was later suspected to have an indolent B-cell lymphoproliferative disorder. Her clinical course highlights both the diagnostic value and the limitations of CrAg testing in non-HIV populations.

## 2. Case Presentation

A 47-year-old Caucasian woman from Ontario, Canada, presented with a two-year history of worsening diplopia, blurry vision, and headaches. She denied fever, weight loss, night sweats, travel history, or immunosuppressive conditions. She was referred to Infectious Diseases in April 2021 for suspected cryptococcal meningitis.

Past medical history included recurrent meningitis (ages 16 and 24), apparent Guillain–Barré syndrome, fibromyalgia, PTSD, Bipolar I disorder, migraine, asthma, and prior hysterectomy. She also had a history of spinal pain treated with lidocaine injections.

On examination, she was alert and oriented with worsening headaches exacerbated by neck rotation and extraocular movement. Neurological, cardiovascular, and respiratory exams were otherwise unremarkable.

An initial lumbar puncture was performed with the collection of 3 mL of CSF per container. Direct microscopic examination of the CSF (liquor) revealed a lymphocytic pleocytosis with no visible yeast forms or organisms and low glucose in a biochemistry test. Her CSF CrAg titer was 1:256. Based on these findings, cryptococcal meningitis was diagnosed, and she was started on liposomal amphotericin B and flucytosine, followed by prolonged fluconazole therapy.

Her HIV tests were repeatedly negative, and other serologies, including HBV, HCV, VZV, rubella, Lyme, *Baylisascaris*, *Aspergillus*, *Blastomyces*, *Coccidioides*, *Histoplasma*, and syphilis, were negative. Autoimmune workup, including bone marrow biopsy, was unrevealing. Table 1 provides the baseline characteristics in detail.

Serial MRIs of the brain, spine, and orbits showed no acute findings or cryptococcal-related lesions. A PET scan showed no FDG-avid nodal or extranodal disease. The only notable imaging finding was mild prominence of the left optic nerve sheath.

Despite completing a 15-month antifungal regimen, she continued to experience headaches, diplopia, falls, and memory loss. Repeat CSF analyses showed persistent lymphocytic pleocytosis and elevated protein. This prolonged course was maintained because the neurologist recommended continuing antifungal therapy while awaiting neuro-oncology assessment. Specifically, she received 400 mg of oral fluconazole daily as maintenance therapy, which she tolerated well without adverse events. CrAg and fungal cultures were repeatedly negative. CSF flow cytometry consistently demonstrated a small monoclonal B-cell population, raising suspicion for a CNS lymphoproliferative process. Table 2 provides the details of CSF flow cytometry testing from multiple lumbar punctures.

Consultations with ophthalmology, hematology, neurology, rheumatology, and immunology yielded no definitive alternative diagnosis. Her case was reviewed at the Regional Cancer Hematology Multidisciplinary Case Conference, which concluded that she likely had an indolent clonal lymphoproliferative process. In December 2023, a ventriculoperitoneal (VP) shunt was placed for symptomatic relief of intracranial hypertension. The patient remains under the care of a neuro-oncologist, with ongoing follow-up in July 2025 that includes regular lumbar punctures for CSF analysis and interval MR head imaging. Despite ongoing monitoring, she continues to experience persistent headaches and visual deficits without significant clinical improvement.

## 3. Discussion

This case presents a diagnostic challenge involving a presumed cryptococcal meningitis based on a single positive CSF CrAg result. Given the lack of response to antifungals, absence of risk factors (e.g., HIV, immunosuppression), and persistently abnormal CSF despite therapy, a false-positive CrAg was considered. Serial testing confirmed negative CSF and serum CrAg, and no fungal organisms grew on repeated cultures.

CSF flow cytometry revealed a persistent monoclonal B-cell population. Though definitive tissue diagnosis was not obtained, the clinical picture was suggestive of a possible primary CNS lymphoma (PCNSL) or low-grade B-cell neoplasm. PCNSL is rare, with ~1500 new cases annually in the U.S. [13].

False-positive CrAg results are documented in multiple contexts. The Cryptococcal Antigen Latex Agglutination System (CALAS) used at this center considers CSF titers ≥1:8 suggestive of infection. However, nonspecific agglutination may occur due to macroglobulins in conditions like SLE, sarcoidosis, cirrhosis, syphilis, scleroderma, gout, and rheumatoid arthritis, per the manufacturer’s datasheet. Pronase pretreatment can mitigate this but is not routine in CSF testing.

Reports in the literature have described multiple mechanisms underlying false-positive or prolonged cryptococcal antigen (CrAg) results. Cross-reactivity with other fungi is a well-recognized cause, with species such as *Trichosporon beigelii*, *Schizophyllum commune*, *Alternaria alternata*, and *Mucor circinelloides* producing polysaccharide antigens that may nonspecifically bind in CrAg assays [9]. Non-infectious conditions associated with increased macroglobulin production, including systemic lupus erythematosus, sarcoidosis, cirrhosis, syphilis, scleroderma, gout, and rheumatoid arthritis have also been implicated in spurious positivity, likely through nonspecific agglutination [10]. Hematologic malignancies represent another important setting for false CrAg reactivity. Isolated CSF CrAg positivity has been observed in patients with lymphoma or other cancers, often without evidence of cryptococcosis [10]. Although most false-positive titers are low (1:2–1:32), rare reports describe higher titers, up to 1:256, in patients with acute myeloid leukemia or hematologic co-morbidity [9,10]. The iatrogenic and technical issues may also contribute. Wilson et al. reported that BBL Port-a-Cul transport vials generated false-positive CrAg results, emphasizing the role of sample handling. Together, these studies underscore that even strongly positive titers may reflect cross-reactivity rather than true cryptococcal infection, particularly in patients with underlying immunoglobulin abnormalities or malignancy [12]. Dubbels and colleagues demonstrated that patients with diffuse large B-cell lymphoma and Waldenström’s macroglobulinemia had low-level CrAg reactivity by lateral flow assay, which reverted to negative with confirmatory latex testing, highlighting assay-dependent variability [11].

In our case, the detection of a persistent monoclonal B-cell population in CSF raises the possibility that paraproteins or immunoglobulin fragments produced by a clonal lymphoproliferative process may have contributed to the initial false-positive CrAg finding.

In this case, it is notable that a strongly positive CrAg titer (1:256) was observed despite negative serum CrAg and cultures. This raises the possibility that macroglobulins or monoclonal immunoglobulins in her CSF cross-reacted with the anti-cryptococcal antibodies in the CALAS test. Agglutination in control samples is typically expected with non-specific reactions, but a fourfold higher titer in the test sample is still considered positive per protocol. Although pronase may reduce nonspecific binding, its use is uncommon in CSF assays.

False-positive cryptococcal antigen (CrAg) results have also been reported in non-traditional clinical specimens, particularly urine. Brito-Santos et al. demonstrated that approximately 30% of fresh urine samples tested positive by lateral flow assay (LFA) despite negative serum results. Importantly, this false positivity was eliminated when urine was preheated at 100 °C for five minutes, yielding 100% concordance with serum [14]. Similarly, Tenforde et al. reported that up to half of urine CrAg LFA positives in HIV-infected patients represented false positives when compared to serum CrAg [15]. Drain et al. further confirmed the limited diagnostic utility of urine-based CrAg testing, finding a positive predictive value as low as 6–13% in clinic-based evaluations. Beyond urine, other factors have been implicated in spurious CrAg reactivity [16]. Rheumatoid factor can cause false-positive serum results, though these can be corrected by boiling or protease treatment [17]. More recently, Liang et al. highlighted multiple contributors to false reactivity including rheumatoid factor, cross-reactive antigens, and technical issues, underscoring that false positivity is not confined to a single specimen type [18]. Together, these reports emphasize that CrAg testing outside of serum and CSF requires cautious interpretation, and that pre-analytical adjustments, such as urine heating, can significantly improve specificity.

Ultimately, the patient’s persistent CSF abnormalities and the presence of a monoclonal B-cell population without systemic disease support the possibility of a CNS-based lymphoproliferative disorder. However, the indolent clinical course and lack of confirmatory imaging suggest an early-stage or atypical presentation. The placement of a VP shunt helped manage her symptoms, though the underlying cause of her meningitis remains incompletely understood. Recent work by Yamashiro-Kanashiro et al. (2025) further demonstrated CrAg reactivity in cases of aspergillosis, histoplasmosis, paracoccidioidomycosis, candidiasis, trichosporonosis, and even bacterial and viral infections, underscoring the assay’s vulnerability to non-specific binding [19].

Comparisons between assay platforms also reveal variability in performance. Schub et al. (2021) found that lateral flow assays (LFAs) and latex agglutination tests may yield discordant results, with LFAs demonstrating greater sensitivity but also an increased risk of cross-reactivity [20]. Wang et al. (2020) reported that low-positive titers in HIV-negative patients, especially in serum and CSF, were often clinically nonspecific and not associated with true cryptococcosis, highlighting the danger of overinterpreting weakly positive results in immunocompetent hosts [21].

Beyond fungal cross-reactivity, autoimmune and rheumatologic conditions may also generate spurious CrAg results. A false-positive CSF CrAg in the context of Libman–Sacks endocarditis has been reported, suggesting that immune complex formation and cross-reactive antigens may interfere with test specificity [22].

Chen et al. (2023) reported a case series of false-positive serum CrAg due to inadequate sample dilution, emphasizing the importance of strict adherence to assay protocols [23]. Conversely, the prozone effect may cause false negatives at high antigen concentrations unless serial dilutions are performed [8].

Taken together, the literature suggests that while CrAg testing is indispensable in HIV-related cryptococcosis, its predictive value is less reliable in immunocompetent or non-HIV patients. Reported false-positive rates are <5% in HIV-infected cohorts but may reach 10–15% in malignancy-associated or immunocompetent settings [7,10,11,21].

## 4. Conclusions

We describe a diagnostically complex case of chronic meningitis in a 47-year-old woman, initially treated for cryptococcal meningitis based on a high CSF CrAg titer. The absence of supportive clinical features, persistent symptoms despite antifungal treatment, negative follow-up CrAg tests, and discovery of a monoclonal B-cell population prompted reconsideration of the diagnosis. The initial CrAg result was ultimately deemed a false positive.

This case illustrates the limitations of CrAg testing and underscores the need for careful interpretation, especially in immunocompetent patients without classic risk factors. While CrAg assays are highly sensitive and specific, false positives due to cross-reactivity, sample handling, or underlying inflammatory/neoplastic states can lead to unnecessary treatment and diagnostic delays.

Persistent monoclonal B-cell populations in CSF, even without radiographic evidence of malignancy, may reflect early or indolent CNS lymphoproliferative processes. Longitudinal follow-up and further studies are needed to clarify their clinical significance. Clinicians should maintain a broad differential diagnosis and integrate laboratory findings with clinical context in evaluating chronic meningitis.

## Figures and Tables

**Table 1 jof-11-00697-t001:** CSF parameters including nucleated cells, lymphocytes, protein, glucose, cryptococcal antigen, CSF culture results and PCR testing. Note: nucleated cells taken from tube 4 as it contained the cell differential.

Date	Nucleated Cells (×10^6^/L)	Lymphocyte (%)	Protein (mg/L)	Glucose (mmol/L)	Cryptococcal Antigen	Bacterial Culture	Fungal Culture	Mycobacterial Culture	HSV/VZV PCR	Other
30 March 2021	27	91	468	3	-	Negative	-	-	-	-
16 April 2021 (2 weeks)	22	86	514	3.1	positive (Titer 1:256)	Negative	-	Negative	-	-
20 April 2021	-	-	-	-	-	Negative	Negative	Negative	Negative	EBV PCR: negative CMV PCR: negative
18 June 2021 (12 weeks)	22	90	514	2.9	Negative	Negative	-	-	-	
8 August 2021 (18 weeks)	12	93	454	2.9	Negative	Negative	Negative	-	-	-
11 August 2021 (18 weeks 2nd LP)	22	93	469	3	Negative	Negative	Negative	-	-	-
23 August 2021 (21 weeks)	20	95	493	2.8	Negative	Negative	Negative	Negative	Negative	16S PCR: negative Tropheryma whipplei PCR: negative Enterovirus PCR: negative
12 October 2021 (28 weeks)	13	94	485	2.9	Negative	Negative	Negative	Negative	Negative	Enterovirus PCR: negative
28 December 2021 (9 months)	24	95	522	3.1	Negative	Negative	-	-	-	-
31 March 2022 (1 year)	19	95	426	2.7	Negative	Negative	Negative	Negative	Negative	-
27 February 2023 (1.5 year)	11	93	426	3.2	Negative	Negative	Negative	Negative	-	-
31 July 2023 (2 years)	23	89	473	3.2	Negative	Negative	Negative	Negative	Negative	EBV PCR: negative Enterovirus PCR: negative
5 October 2023 (2.5 years)	15	96	466	3.4	Negative	Negative	-	-	-	-
30 October 2023 (2.5 years 2nd LP)	11	97	464	3.2	Negative	Negative	Negative	-	-	-

**Table 2 jof-11-00697-t002:** CSF flow cytometry testing from multiple lumbar punctures.

Date	Flow Cytometry
31 July 2023	Monoclonal B-cell population representing approximately 3% of the total leukocyte count positive for CD19, dim CD20, and expressing kappa light chains. Suspicious for a B-cell lymphoproliferative neoplasm
17 August 2023	Not suggestive of a B-cell lymphoproliferative neoplasm
5 October 2023	Monoclonal B-cell population representing approximately 2% of the total leukocyte count positive for CD19, CD20 and expressing dim kappa light chains. Suspicious for a B-cell lymphoproliferative neoplasm
30 October 2023	A population of B cells was detected but is below the level of enumeration, is positive for CD19, CD20 and expressing dim kappa light chains. Uncertain clinical significance

## Data Availability

No new data were created or analyzed in this study. Data sharing is not applicable to this article.

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
