# Peer review of "The Diagnostic Challenge of a False-Positive Cryptococcal Antigen in Chronic Meningitis with Suspected Indolent CNS B-Cell Lymphoproliferative Neoplasm"

_jof, 2025, doi:10.3390/jof11100697_

Round 1

Reviewer 1 Report

Comments and Suggestions for Authors

Dear Editor,

Thank you for the invitation to review the article titled (The Diagnostic Challenge of a False-Positive Cryptococcal Antigen in Chronic Meningitis with Suspected Indolent CNS B-Cell Lymphoproliferative Neoplasm)

The case report is important and well-founded. Cryptococcosis is a disease of global importance and should be investigated, even in areas of high epidemic proportions. Canada is indeed part of the global cryptococcal scenario, including cryptococcosis caused by C. gattii. I have some considerations.

First, include possible reports of cross-reactions related to prolonged 

Second, include an article on false positives in other clinical specimens (URINE and others).

This information is important to elucidate the possible false positives present in the Cryptococcal antigen research technique.

Best regards

Author Response

Comment 1:  include possible reports of cross-reactions related to prolonged 

Reports in the literature have described multiple mechanisms underlying false-positive or prolonged cryptococcal antigen (CrAg) results. Cross-reactivity with other fungi is a well-recognized cause, with species such as Trichosporon beigelii, Schizophyllum commune, Alternaria alternata, and Mucor circinelloides producing polysaccharide antigens that may nonspecifically bind in CrAg assays.(12) Non-infectious conditions associated with increased macroglobulin production, including systemic lupus erythematosus, sarcoidosis, cirrhosis, syphilis, scleroderma, gout, and rheumatoid arthritis have also been implicated in spurious positivity, likely through nonspecific agglutination.(13) Hematologic malignancies represent another important setting for false CrAg reactivity. Isolated CSF CrAg positivity has been observed in patients with lymphoma or other cancers, often without evidence of cryptococcosis.(13) Although most false-positive titers are low (1:2–1:32), rare reports describe higher titers, up to 1:256, in patients with acute myeloid leukemia or hematologic co-morbidity.(12,13) The iatrogenic and technical issues may also contribute. Wilson et al. reported that BBL Port-a-Cul transport vials generated false-positive CrAg results, emphasizing the role of sample handling. Together, these studies underscore that even strongly positive titers may reflect cross-reactivity rather than true cryptococcal infection, particularly in patients with underlying immunoglobulin abnormalities or malignancy.(14) Dubbels and colleagues demonstrated that patients with diffuse large B-cell lymphoma and Waldenström’s macroglobulinemia had low-level CrAg reactivity by lateral flow assay, which reverted to negative with confirmatory latex testing, highlighting assay-dependent variability. (15) In our case, the detection of a persistent monoclonal B-cell population in CSF raises the possibility that paraproteins or immunoglobulin fragments produced by a clonal lymphoproliferative process may have contributed to the initial false-positive CrAg finding. 

Comment 2:  include an article on false positives in other clinical specimens (URINE and others).

False-positive cryptococcal antigen (CrAg) results have also been reported in non-traditional clinical specimens, particularly urine. Brito-Santos et al. demonstrated that approximately 30% of fresh urine samples tested positive by lateral flow assay (LFA) despite negative serum results. Importantly, this false positivity was eliminated when urine was preheated at 100 °C for five minutes, yielding 100% concordance with serum.(16) Similarly, Tenforde et al. reported that up to half of urine CrAg LFA positives in HIV-infected patients represented false positives when compared to serum CrAg.(17) Drain et al. further confirmed the limited diagnostic utility of urine-based CrAg testing, finding a positive predictive value as low as 6–13% in clinic-based evaluations. Beyond urine, other factors have been implicated in spurious CrAg reactivity. (18) Rheumatoid factor can cause false-positive serum results, though these can be corrected by boiling or protease treatment. (19) More recently, Liang et al. highlighted multiple contributors to false reactivity including rheumatoid factor, cross-reactive antigens, and technical issues, underscoring that false positivity is not confined to a single specimen type.(20) Together, these reports emphasize that CrAg testing outside of serum and CSF requires cautious interpretation, and that pre-analytical adjustments, such as urine heating, can significantly improve specificity.

Reviewer 2 Report

Comments and Suggestions for Authors

Thank you for submitting this interesting case report.

The manuscript addresses an unusual diagnostic challenge involving a presumed false-positive cryptococcal antigen result in the setting of chronic meningitis. While the topic is potentially valuable, the current version requires substantial revision before it can be considered for publication. Below, I outline the main concerns, organised by manuscript section.

Introduction section:

  • Please add a concise review of the known causes of false-positive results (cross-reactivity, interference by immunoglobulins, prozone effect, technical errors) and their reported frequency in the literature.

  • It would also be helpful to contrast the high reliability of CrAg in HIV-infected patients with the more ambiguous significance in immunocompetent or non-HIV populations, which is more relevant to your case.

  • The introduction starts in medias res. It should be better to start with the introduction of meningitis, then cronical meningitis, the causes and then the Criptococcus as a cause of meningitis, then diagnosis and differential diagnosis and report, here, the HIV vs non-HIV subject, that is more pertinent to your case
  • Move to the final section of the introduction of your case presentation, now in the first lines

Case report section:

  • Specify the ethnicity
  • How did you see the direct observation of liquor at the microscope? 
  • Why 15 months of treatment without any proven diagnosis? report the regimens, exact dose and descalation, and any adverse events
  • Add follow-up until 2025
  • add the volume of liquor collected

Discussion

This section is not uniform and should be improved. I suggest making the text more fluent, adding references and improving the confutation of your case. I suggest reading and citing:

  • Yamashiro-Kanashiro, E. H., Kanunfre, K. A., Mimicos, E. V., de Freitas, V. L. T., Rocha, M. C., Shimoda Nakanishi, É. Y., Miyachi, M. E., Batista, M. V., Martinez, R., Litvoc, M. N., Sumita, N. M., Fonseca, C. A., Rodrigues, H. G., Lagonegro, E. R., & Shikanai Yasuda, M. A. (2025). "Reactivity of cryptococcal lateral flow assay in aspergillosis, histoplasmosis, paracoccidioidomycosis, candidiasis, trichosporonosis, bacterial, and viral infections". Medical mycology63(8), myaf068. https://doi.org/10.1093/mmy/myaf068
  • Schub, T., Forster, J., Suerbaum, S., Wagener, J., & Dichtl, K. (2021). Comparison of a Lateral Flow Assay and a Latex Agglutination Test for the Diagnosis of Cryptococcus Neoformans Infection. Current microbiology78(11), 3989–3995. https://doi.org/10.1007/s00284-021-02664-w
  • Wang, X., Cheng, J. H., Zhou, L. H., Zhu, J. H., Wang, R. Y., Zhao, H. Z., Jiang, Y. K., Huang, L. P., Yip, C. W., Que, C. X., Zhu, M., & Zhu, L. P. (2020). Evaluation of low cryptococcal antigen titer as determined by the lateral flow assay in serum and cerebrospinal fluid among HIV-negative patients: a retrospective diagnostic accuracy study. IMA fungus11, 6. https://doi.org/10.1186/s43008-020-00028-w
  • Isseh, I. N., Bourgi, K., Nakhle, A., Ali, M., & Zervos, M. J. (2016). False-positive cerebrospinal fluid cryptococcus antigen in Libman-Sacks endocarditis. Infection44(6), 803–805. https://doi.org/10.1007/s15010-016-0909-8
  • Chen, W. Y., Zhong, C., Zhou, J. Y., & Zhou, H. (2023). False positive detection of serum cryptococcal antigens due to insufficient sample dilution: A case series. World journal of clinical cases11(8), 1837–1846. https://doi.org/10.12998/wjcc.v11.i8.1837

Comments on the Quality of English Language

The English could be improved to more clearly express the research.

Author Response

Comment 1:  Introduction section:

  • Please add a concise review of the known causes of false-positive results (cross-reactivity, interference by immunoglobulins, prozone effect, technical errors) and their reported frequency in the literature.

  • It would also be helpful to contrast the high reliability of CrAg in HIV-infected patients with the more ambiguous significance in immunocompetent or non-HIV populations, which is more relevant to your case.

  • The introduction starts in medias res. It should be better to start with the introduction of meningitis, then cronical meningitis, the causes and then the Criptococcus as a cause of meningitis, then diagnosis and differential diagnosis and report, here, the HIV vs non-HIV subject, that is more pertinent to your case
  • Move to the final section of the introduction of your case presentation, now in the first lines

Meningitis is a potentially life-threatening syndrome characterized by inflammation of the meninges, which may present acutely or chronically depending on the underlying etiology. Acute meningitis is most often viral or bacterial, while chronic meningitis defined as clinical meningitis with cerebrospinal fluid (CSF) pleocytosis lasting more than four weeks has a broad differential diagnosis (1,2). Causes include infections (fungal, mycobacterial, parasitic, atypical bacterial), inflammatory and autoimmune disorders (neurosarcoidosis, CNS vasculitis, IgG4-related disease, systemic lupus erythematosus), and malignancy (1,2). Among infectious causes, Cryptococcus species are the most common fungal pathogens leading to chronic meningitis. Cryptococcus neoformans primarily affects immunocompromised hosts, particularly those with advanced HIV infection, while Cryptococcus gattii can also cause disease in immunocompetent individuals (3,4,5). Diagnosis relies on clinical suspicion, CSF findings, culture, and cryptococcal antigen (CrAg) testing. CrAg assays including latex agglutination, enzyme immunoassay, and lateral flow assays are highly sensitive and specific, and their rapid turnaround has made them indispensable for early recognition of cryptococcosis. In HIV-infected populations, CrAg testing has shown excellent diagnostic performance, with sensitivity and specificity exceeding 98% in both serum and CSF (8). In this setting, a positive CrAg is highly predictive of true infection and typically warrants immediate antifungal therapy. By contrast, in immunocompetent or non-HIV patients, the interpretation of CrAg positivity is more challenging because the pretest probability of cryptococcosis is lower and the differential diagnosis broader (9,10). False-positive CrAg results, although uncommon, have been described in multiple contexts. Cross-reactivity with other fungi, such as Trichosporon beigelii, Schizophyllum commune, Alternaria alternata, and Mucor circinelloides, has been reported (12). Interference from immunoglobulins, particularly macroglobulins and rheumatoid factor, can produce nonspecific agglutination, with cases described in systemic lupus erythematosus, rheumatoid arthritis, and hematologic malignancies (13,15). High titers have occasionally been documented in the absence of cryptococcal infection, particularly in patients with hematologic disease (13). Technical issues, including contamination and cross-reactivity with specimen transport media, have also been implicated (14). In addition, the prozone effect where excessive antigen concentrations inhibit visible agglutination can result in paradoxical false negatives unless serial dilutions are performed (10). Reported frequencies of false positivity are generally low in HIV-infected patients but appear higher in immunocompetent individuals or those with alternative pathology. This variability underscores the importance of interpreting CrAg results in clinical context. In the case we present, an immunocompetent woman with chronic meningitis was initially diagnosed with cryptococcal meningitis on the basis of a high CSF CrAg titer but was later suspected to have an indolent B-cell lymphoproliferative disorder. Her clinical course highlights both the diagnostic value and the limitations of CrAg testing in non-HIV populations.

Comment 2:  Case report section:

  • Specify the ethnicity: A 47-year-old Caucasian woman from Ontario, Canada
  • How did you see the direct observation of liquor at the microscope? add the volume of liquor collected: Lumbar puncture was performed with collection of 3 mL of CSF per container. Direct microscopic examination of the CSF (liquor) revealed a lymphocytic pleocytosis with no visible yeast forms or organisms.
  • Why 15 months of treatment without any proven diagnosis? report the regimens, exact dose and descalation, and any adverse events: This prolonged course was maintained because the neurologist recommended continuing antifungal therapy while awaiting neuro-oncology assessment. Specifically, she received oral fluconazole 400 mg daily as maintenance therapy, which she tolerated well without adverse events.
  • Add follow-up until 2025: The patient remains under the care of a neuro-oncologist, with ongoing follow-up in July 2025 that includes regular lumbar punctures for CSF analysis and interval MR head imaging. Despite ongoing monitoring, she continues to experience persistent headaches and visual deficits without significant clinical improvement.

Comment 3: Discussion section:

This section is not uniform and should be improved. I suggest making the text more fluent, adding references and improving the confutation of your case. I suggest reading and citing. 

Recent work by Yamashiro-Kanashiro et al. (2025) further demonstrated CrAg reactivity in cases of aspergillosis, histoplasmosis, paracoccidioidomycosis, candidiasis, trichosporonosis, and even bacterial and viral infections, underscoring the assay’s vulnerability to non-specific binding (21).

Comparisons between assay platforms also reveal variability in performance. Schub et al. (2021) found that lateral flow assays (LFAs) and latex agglutination tests may yield discordant results, with LFAs demonstrating greater sensitivity but also an increased risk of cross-reactivity (22). Wang et al. (2020) reported that low-positive titers in HIV-negative patients, especially in serum and CSF, were often clinically nonspecific and not associated with true cryptococcosis, highlighting the danger of overinterpreting weakly positive results in immunocompetent hosts (23).

Beyond fungal cross-reactivity, autoimmune and rheumatologic conditions may also generate spurious CrAg results. A false-positive CSF CrAg in the context of Libman–Sacks endocarditis has been reported, suggesting that immune complex formation and cross-reactive antigens may interfere with test specificity (24).

Chen et al. (2023) reported a case series of false-positive serum CrAg due to inadequate sample dilution, emphasizing the importance of strict adherence to assay protocols (25). Conversely, the prozone effect may cause false negatives at high antigen concentrations unless serial dilutions are performed (10).

Taken together, the literature suggests that while CrAg testing is indispensable in HIV-related cryptococcosis, its predictive value is less reliable in immunocompetent or non-HIV patients. Reported false-positive rates are <5% in HIV-infected cohorts but may reach 10–15% in malignancy-associated or immunocompetent settings (12,13,15,23).

Round 2

Reviewer 2 Report

Comments and Suggestions for Authors

Dear Authors,

All the points have been addressed, and the paper has been improved.